# Human Evolution and Dietary Ethanol

**DOI:** 10.3390/nu13072419

**Published:** 2021-07-15

**Authors:** Robert Dudley, Aleksey Maro

**Affiliations:** 1Department of Integrative Biology, University of California, Berkeley, CA 94720, USA; alekseymaro@berkeley.edu; 2Smithsonian Tropical Research Institute, P.O. Box 2072, Balboa, Panama

**Keywords:** alcoholism, evolution, fermentation, frugivory, *Homo*, primate, yeast

## Abstract

The “drunken monkey” hypothesis posits that attraction to ethanol derives from an evolutionary linkage among the sugars of ripe fruit, associated alcoholic fermentation by yeast, and ensuing consumption by human ancestors. First proposed in 2000, this concept has received increasing attention from the fields of animal sensory biology, primate foraging behavior, and molecular evolution. We undertook a review of English language citations subsequent to publication of the original paper and assessed research trends and future directions relative to natural dietary ethanol exposure in primates and other animals. Two major empirical themes emerge: attraction to and consumption of fermenting fruits (and nectar) by numerous vertebrates and invertebrates (e.g., *Drosophila* flies), and genomic evidence for natural selection consistent with sustained exposure to dietary ethanol in diverse taxa (including hominids and the genus *Homo*) over tens of millions of years. We also describe our current field studies in Uganda of ethanol content within fruits consumed by free-ranging chimpanzees, which suggest chronic low-level exposure to this psychoactive molecule in our closest living relatives.

## 1. Introduction

The argument of the “drunken monkey” hypothesis is that alcohol (and primarily the ethanol molecule) is a low-level but routine component of the diet for all animals that consume fruits and nectar [1,2]. In addition to providing a useful long-distance olfactory cue to localize nutritional resources and to identify ripe and calorically rich fruits up close, ethanol may also act as a feeding stimulant (as in modern humans, via the well-studied aperitif effect; [3]). Humans first began intentional fermentation during the Meso-Neolithic transitional period broadly coincident with the domestication of crops, and ethanol consumption has correspondingly been viewed as a fairly recent phenomenon relative to the origin of our species. However, dietary consumption of ethanol likely characterizes all frugivorous and nectarivorous animals, including primates and the hominoid lineage leading to modern humans. Millions of years of interaction among flowering plants, fermentative yeast, and numerous vertebrate lineages thus suggest a linkage between ethanol ingestion and acquisition of nutritional reward. We also see in diverse animal taxa, as well as in modern humans, substantial genetic variation in the ability to metabolize ethanol that is consistent with natural selection to this end.

The natural role of ethanol in animal nutrition has been largely underestimated in the zoological literature. For example, ethanol in ripe and fermenting fruits has been proposed to be largely aversive to vertebrate consumers [4]. More recently, information from behavioral, ecological, and genomic studies indicates an impressive commonality of behavioral and physiological responses to ethanol, and in taxa ranging from fruit flies to primates. The overarching concept that unites these studies is evolution, which can sometimes provide novel insights into questions of human health and behavior [5,6]. Here, we review advances in the field of comparative ethanol biology since the first publication of the “drunken monkey” hypothesis [1], and describe emerging themes in the 140 English language citations to the original paper (Google Scholar; January 2001–April 2021). We also provide preliminary information on ethanol content of fruits consumed in nature by our nearest living relatives, the chimpanzees. Given that chimpanzees mostly eat ripe fruits (e.g., up to 86% of the time; [7,8]), and that a comparable diet is thought to have pertained to the earliest hominins [9,10,11], these data suggest that low-level ethanol ingestion was an important feature of human nutrition over evolutionary time. Such ethanol consumption via frugivory could, in turn, have resulted in physiological and sensory adaptations that, today, yield hedonic reward following dietary exposure to this molecule [2]. Predictions of the “drunken monkey” hypothesis and relevant empirical findings since 2000 are provided in Table 1.

## 2. Vertebrate Responses to Naturally Occurring Ethanol

Sugars within ripe and over-ripe fruits serve as caloric motivation for consumption by animals, primarily mammals and birds, that subsequently disperse the seeds. Ripe fruits must be attractive to these consumers and must also present sufficient nutritional reward so as to elicit consumption. However, the ubiquity of yeasts in natural environments indicates the potential for fermentation prior to consumption by vertebrates [34,35]. Anaerobic fermentation by yeasts and ethanol generation have been dated using molecular methods to coincide with the origin of fleshy and sugar-rich fruits in the Cretaceous period [36] and may specifically have evolved to inhibit activity of bacterial competitors within fruit pulp [37]. Fruit decomposition can thus be viewed as a race in time between microbes and dispersal agents, and correspondingly, there is selection on vertebrate sensory mechanisms to facilitate rapid localization and consumption of these transient resources.

Fermentation of fruit crops is most pronounced in warm, humid environments such as tropical rainforests, the habitat of most frugivorous primates today. For example, ripe palm fruits (*Astrocaryum standleyanum*) contain ~0.6% ethanol within the pulp, but over-ripe fruits have much higher levels, averaging 4.5% [12]. Substantial levels of ethanol within pulp also characterize fruits in Southeast Asia over a range of ripening stages [13]. Animals consuming these fruits will necessarily ingest ethanol at low concentrations. Given that animal frugivores can consume 5%–10% of their body weight daily in ripe fruit, even the aforementioned low concentrations will yield substantial chronic dosage. Floral nectars in the tropics can also ferment and yield substantial ethanol concentrations. Wild tree shrews and slow lorises feed from palm flowers (*Eugeissona tristis*) in Malaysia that contain significant levels of ethanol within the nectar [14]. Although the animals never become overtly inebriated, hair samples contain high levels of a secondary metabolite of ethanol (ethyl glucuronide), consistent with high chronic exposure. Laboratory choice trials with two species of nectar-feeding primates indicate increasing preference for higher-concentration ethanol solutions [18] (see also [19] for analogous experiments with a primate frugivore). Additionally, wild chimpanzees consume anthropogenically sourced fermentations of palm sap within the tree canopy, at least at one site in West Africa [20]. Critically, the assertion that ethanol is toxic and renders fruit unpalatable to vertebrates [4] has been empirically falsified for mammalian dispersal agents [21].

In tropical rainforests, ripe fruit is a transient and spatially heterogeneous resource. Olfactory plumes of ethanol provide, however, an honest signal of caloric availability to potential consumers downwind. The olfactory sensitivity of primates to various alcohols, including ethanol, is well-developed [16,17], but this sensory capacity has not been demonstrated under field conditions. Adult fruit flies, however, use ethanol plumes to locate suitable oviposition sites on ripe fruit. The study of ethanol responses in *Drosophila* now represents a useful model system for understanding molecular pathways of inebriation in humans [38]. Additionally, behavioral preferences by fruit flies for ethanol-containing substrates are correlated with the ability to metabolize ethanol, suggesting a direct link between metabolic capacity and sensory attraction [39]. Similarly, ethanol is not aversive to fruit-feeding birds and bats [22,23] and is sometimes consumed at lethal levels [24,25]. In rodents, ethanol evokes neural hyperactivity in brain-feeding circuits, further supporting evolutionary associations between consumption of fermented substrates and caloric gain [40]. Most importantly, a recent survey of wild primate diets [15] demonstrated the widespread consumption of fruits in the late stages of fermentation (as assessed by human observers). Because ethanol may be present within ripe fruits with no obvious external signs of microbial activity, this study provides a conservative estimate of actual dietary exposure; a quantitative assessment of ethanol concentrations within consumed fruits across the entire spectrum of palatability is clearly now called for.

## 3. Evolutionary Consequences of Dietary Ethanol

If chronic dietary exposure to ethanol inevitably derives from frugivory (and from nectarivory), then selection will favor the evolution of metabolic adaptations that maximize physiological benefits but minimize costs of exposure. Higher concentrations of ethanol may, by contrast, be stressful and cause harm. Such a nonlinear dose-response curve is termed hormesis and is an evolutionary outcome that increases overall organismal fitness given natural exposure to various compounds at low concentrations [41,42,43]. A key prediction of the “drunken monkey” hypothesis, therefore, is that hormetic benefits will pertain to animals at low, naturally occurring levels of ethanol exposure.

In support of this claim, longevity (as well as female fecundity) of fruit flies is increased at low atmospheric concentrations of ethanol but decreases at zero exposure and at higher concentrations [28,29,30]. Laboratory rodents similarly show decreased mortality at intermediate levels of ethanol ingestion [31]. In humans, epidemiological studies suggest a reduction in cardiovascular risk and overall mortality at low levels of ethanol consumption relative either to abstinence or to higher levels of intake [32,33]. Consequences of chronic ethanol ingestion for human reproductive fitness have not been evaluated, but we might expect a similar outcome as with longevity. No current data address the hormetic effects of ethanol on wild animals with variable levels of dietary availability, but logistically, such long-term measurements can be carried out in appropriate contexts (e.g., field-banded tracking of individual hummingbirds through their lifespan at different sites with variable extent of nectar fermentation).

Evolutionary arguments also predict that intra- and interspecific variation in the ability to metabolize ethanol will correspond to its relative dietary inclusion. Alcohol dehydrogenase (ADH) initially converts ethanol to acetaldehyde, which then is acted upon by aldehyde dehydrogenase (ALDH) to yield acetate used for energy yield in oxidative pathways. Both ADH and ALDH exist in a number of different allelic forms characterized by varying catalytic efficiencies, which in *Drosophila* flies are well-known to correlate with natural levels of environmental ethanol exposure [2,39]. Furthermore, in the lineage of great apes that led to modern humans, there is a pronounced genetic signature demonstrating comparable evolutionary responses to chronic dietary exposure to ethanol. Paleogenetic reconstruction of alcohol dehydrogenase genes across the hominid phylogeny indicates a dramatically enhanced catabolic capacity in one particular ADH (ADH4, as encoded by the *ADH7* allele), starting at about 10 Mya [26]. ADH4, although only one of multiple ADH forms present in mammals, is found primarily in the mouth and digestive tract and thus effects the “first pass” at the digestion of ethanol. This enzyme became dramatically better at metabolizing ethanol following the phyletic split between the lineage leading to modern orangutans and to the other great apes, including ourselves. It thus correlates well with increasing terrestrialization among the African apes, possibly yielding greater access to fermenting fruit crops on the ground, and thus resulting in increased ethanol within the diet [26]. The same mutation also characterizes ADH4 of the Madagascan aye-aye, which routinely feeds on nectar from flowers of an endemic palm. Although ethanol content is not characterized for such flowers, studies with captive aye-ayes demonstrate a preference for consumption of low-level ethanol within sugar solutions [18].

Moreover, a recent study [27] evaluated variation in ADH 4 across 79 mammal species; multiple losses of function in *ADH7* (i.e., pseudogenization) and relaxed selection on this allele were found for those taxa with little or no presumed dietary exposure to ethanol (e.g., whales). Contrariwise, natural selection was apparently intensified on *ADH7* for those species specializing on either fruit or nectar [27]. Although the actual extent of dietary ethanol consumption is not known for the study species, clearly the likelihood of its chronic ingestion must be higher for frugivores and nectarivores. Quantitative specification of ethanol exposure, in conjunction with assessment of genetic changes in the other ethanol-metabolizing enzymes (e.g., ADH1, ADH2, and numerous ALDH polymorphisms) is now called for to assess the overall evolutionary response to fermented nutritional substrates. 

Hominoids (i.e., the lesser and greater apes) also exhibit an evolutionary loss of uricase as a consequence of accumulating deleterious mutations in the corresponding gene (starting ~20 Mya; [44,45]). Modern humans correspondingly exhibit very high blood levels of uric acid and show amplification of fat accumulation (and of the metabolic syndrome more generally) given chronic fructose ingestion [46,47,48]. Ethanol consumption also stimulates fructose production by the liver, as well as more widespread production of uric acid, with both effects acting synergistically to increase overall fat storage [49,50]. The psychoactive and hedonic properties of ethanol and fructose are also similar, facilitating addictive responses to these naturally occurring compounds within fruit [51]. Such changes in both the uricase gene and in genes directly involved in ethanol catabolism are consistent with positive selection on dietary preference for fruit sugars and their fermentation products and are possibly linked with sensory mechanisms facilitating their efficient consumption and digestion.

In addition to aforementioned interspecific studies of ethanol metabolism, there is also substantial intraspecific genetic variation in physiological responses to ethanol, at least among modern human populations. In particular, slow-acting ALDH occurs at high frequencies in East Asian humans, and yields toxic acetaldehyde buildup following the consumption of ethanol [52,53]. Such variation, in turn, has been correlated with the propensity towards alcoholism for certain populations. Rates of alcoholism, however constructed definitionally, tend to be much lower within East Asian populations, consistent with the deterrent effects of elevated acetaldehyde [54,55]. Although genotype-by-environment interactions are also likely to be involved, the interacting dynamics of ethanol catabolism and accumulation of the acetaldehyde intermediate product are apparently protective against excessive alcohol consumption [56].

Finally, multigenerational exposure to high levels of dietary ethanol can result in significant changes to the gut microbiome, at least in laboratory rodents [57]. This intriguing outcome, mediated either directly by ethanol or by its downstream metabolic products, may also indicate systemic neural regulation of ingestion as influenced by endogenous gut fauna. The role of the microbiome in mediating physiological and behavioral responses to ethanol, either across the lifespan or in evolutionary time, has never been evaluated for free-ranging vertebrates, but clearly is of adaptive relevance. As with aforementioned molecular evolutionary studies of ADH and ALDH, comparative studies of the gut microbiome among frugivorous and nectarivorous species (and including birds as well as mammals) would elucidate correlates of microbical community composition relative to chronic ethanol ingestion and may indicate a role for selection in promoting higher rates of ethanol consumption so as to increase energetic gain while feeding.

## 4. Natural Ethanol Exposure in Chimpanzees

Recent field studies of chimpanzee-consumed fruits in Uganda suggest a chronic low-level ingestion of ethanol, albeit at sub-inebriating levels that are nonetheless consistent with physiological consequences. The Ngogo population of Eastern chimpanzees (*P. troglodytes schweinfurthii*) in Kibale National Park reside in a forest with a low density of a high-output, asynchronously fruiting fig species (*Ficus mucuso*), which is consumed preferentially more than any other fruit (i.e., 18%–34% of total feeding time; [58,59]). In 2019 and 2020, we determined ethanol concentrations for *F. mucuso* fruits as well as for a diversity of other consumed fruit species. By visiting *F. mucuso* trees with chimpanzees actively foraging in the canopy, we could collect ripe figs either immediately after they fell, following disturbance or by rejection, or within an hour of having fallen (as evidenced by wet latex at the stem). We also collected unripe *F. mucuso* during part of the field season, which the chimpanzees eat during periods of food shortage. Collected figs were frozen at the field station to arrest fermentation. We determined ethanol concentrations within individual fruits using an infrared gas analyzer on homogenized pulp samples, and also via ethanol vapor measurements in the headspace over pulp samples. Prior to these measurements, for each fruit we also assessed its mass, puncture resistance, sugar concentration, surface reflectance, and presence or absence of fig wasps, so as to correlate quantitatively ethanol content with stages of ripeness, and to assess which factors most influence microbial ethanol production. Data obtained to date indicate ethanol levels within ripe figs ranging from negligible amounts to as high as several percent (weight of ethanol/weight of fruit), consistent with values determined for other primate-consumed fruits [12,13].

Levels of ethanol consumption are determined both by ingested food volume and by intrinsic concentration. A typical daily consumption of ~6 kg of fruit at an ethanol content of only 0.23% would correspond to ingestion of one standard drink (i.e., 14 g of ethanol in the USA). Moreover, adult Eastern chimpanzees in the wild weigh substantially less than humans (i.e., only 30–40 kg; [60]), suggesting a much higher body-mass specific exposure. If consumed fruit were to contain 1% ethanol on average, then consuming 6 kg of fruit daily would yield >4 standard drinks daily, and a much higher body-mass specific rate of exposure. These preliminary calculations suggest that ethanol ingestion via frugivory is non-trivial in wild chimpanzees, and can easily approach chronic exposure of physiological relevance, if not of occasional inebriation.

Further assessment of dosage via dietary ingestion would require knowledge of rates of ethanol absorption and catabolism in chimpanzees, which are not necessarily the same as those in humans. Enzymatic activity relative to ethanol degradation is variable among mammalian taxa, and even among modern human populations [56,61,62]. Direct measurement of blood-ethanol levels in free-ranging chimpanzees and other frugivores would be informative in this regard. Nonetheless, chimpanzees may have evolved specific behavioral and physiological responses to ethanol commensurate with its natural occurrence within ripe and over-ripe fruit. Ripening in figs poses particular challenges to frugivores in that overt and substantial color changes otherwise indicating suitability of fruits for consumption are not present in this genus (*Ficus*: Moraceae). Both short-range olfactory and tactile cues are thus more important in identifying ripe fruits, with reduced use of visual cues [63]. Equally relevant to foraging outcomes are features of spatial and temporal heterogeneity in fruit ripening and fermentation. For example, fruits growing within the same tree may be vertically stratified, with fruits higher in the canopy being larger and containing more sugars [64] and possibly higher ethanol content as well.

Moreover, fig wasps are the mutualistic tenants and obligate pollinators of figs and may influence fermentation outcomes via different microbiota that they vector into fruits. Fig wasp behavior and ecology are highly variable among species, as are the chemical and structural features of different figs, which may, in turn, influence outcomes of microbial colonization and growth. Finally, endogenous fermentation of sugars is likely to vary with local climate, and in particular with average ambient temperatures. Lower elevations likely yield fruits with higher ethanol concentrations, given faster yeast growth in hotter climates. Our data for Ngogo (situated at 1400 m above sea level) likely represent conservative values for fruit-ethanol concentrations relative to those within lowland tropical rainforests where most frugivorous animals are found. Nonetheless, these preliminary measurements suggest sustained exposure of our closest living relatives to dietary ethanol and establish a methodological framework for further investigation into the natural consumption of fermented fruits.

## 5. Conclusions

Behavioral responses to naturally occurring ethanol can be advantageous for many animals, may be ancestral in primates, and have substantial implications for modern humans relative to both benign consumption of alcohol and excessive levels of drinking. A number of empirical questions can be posed to further assess the generality of these evolutionary arguments. In natural ecosystems, how do animals localize fermenting nutritional resources, and what are typical blood-ethanol levels within frugivores and nectarivores? Do the hormetic effects of low-level ethanol consumption extend more generally to all species exposed to this molecule over evolutionary time? Are there particular sensory mechanisms in some species that predispose them to excessive ethanol ingestion under artificial conditions of high supply? For example, ethanol evokes hyperactivity in the brain-feeding circuits of rodents, consistent with a general role as an appetitive stimulant [40]. Fermentation by yeasts of simple carbohydrate substrates is widespread in terrestrial environments, yet the natural background of ethanol availability has been largely ignored by biologists and clinicians alike, nutritional and health implications for modern humans notwithstanding. We therefore encourage further studies of ethanol-seeking activities in the natural world, as such behaviors (and their underlying genetic underpinnings) may yield novel insights into contemporary human consumption and misuse of alcoholic beverages.

## Figures and Tables

**Table 1 nutrients-13-02419-t001:** Predictions of the “drunken monkey” hypothesis and supporting empirical evidence.

Prediction	Supporting Evidence	References
Ethanol occurs naturally at low levels within many fruits and nectars.	A variety of tropical fruits, as well as some nectars, contain ethanol at low concentrations.	[12,13,14,15]
Olfaction can be used to localize and preferentially select ethanol-containing nutritional resources.	Fruits consumed by primates produce numerous volatiles, including ethanol. Olfactory abilities are well-developed in primates, but have not been explicitly tested relative to use in fruit localization or selection.	[16,17]
Ethanol at low concentrations is not aversive to frugivores and nectarivores.	Diverse vertebrates consume food items containing low-concentration ethanol.	[18,19,20,21,22,23,24,25]
Ethanol acts as a feeding stimulant.	Modern humans increase caloric ingestion following consumption of an aperitif. Effects of dietary ethanol on ingestion rates for free-ranging primates have not yet been evaluated.	[3]
Genetic variation in the ability to metabolize ethanol is correlated with the extent of dietary exposure.	Substantial variation in ADH tracks dietary inclusion of fruit and nectar among mammals. Ethanol catabolism was up-regulated in African apes ~10 Mya ago, in parallel with terrestrialization.	[26,27]
Hormetic advantage derives from chronic consumption of ethanol.	Mortality is reduced at low levels of ethanol ingestion in modern humans and rodents, and also in *Drosophila* flies exposed to low-concentration ethanol vapor.	[28,29,30,31,32,33]

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
