# Peer review of "Human Evolution and Dietary Ethanol"

_nutrients, 2021, doi:10.3390/nu13072419_

Round 1
Reviewer 1 Report
It is certainly a topic of great interest. The bibliography is perfectly selected and the authors are great experts on this subject. This reviewer just has a few comments and questions:
- Although the authors cite the works of Carriegan et al. PNAS 2014 and Janiak et al. Biol. Letters, 2020; discussion of this information seems superficial. I recommend discussing in greater depth the importance of the genetic study of ADH to understand the evolution of alcohol consumption in primates. They should delve into relevant aspects such as: the psedogenization of ADH7, the activity of ADH4, polymorphisms ... In the manuscript this information appears scattered and must be integrated.
- A summary table of the scientific evidence in favor of the drunken monkey hypothesis would be very helpful to the reader.
- A graphical abstract is mandatory in a good review.
Author Response
Responses to Reviewer 1:
1. We have reorganized and better integrated this discussion, and have expanded assessment of the evolution of ADH and ALDH genes involved in ethanol catabolism.
2. A summary table is now included.
3. Graphical abstracts are not used by Nutrients.
Reviewer 2 Report
I enjoyed reading this manuscript and I appreciate its effort to move the frontier. I support publication and I offer only a few minor comments.
1. Page 2, line 54 –– You could say “as much as 86% of the time” to quantify your meaning of “most”. There may be higher annual estimates elsewhere in the literature, but for now I would steer you to Wrangham et al. 1996:
Wrangham, R. W., C. A. Chapman, A. P. Clark-Arcadi, and G. Isabirye-Basuta. 1996. Social ecology of Kanyawara chimpanzees: implications for understanding the costs of great ape groups, Pages 45-57 in W. C. McGrew, L. F. Marchant, and T. Nishida, eds. The great ape societies. Cambridge, Cambridge University Press.
2. Page 2, line 55 –– “early humans”. I think many anthropologists would reserve the word human for members of the genus Homo. Better to say “the earliest hominins” instead
3. Page 2, line 91 –– you could also cite the paper by Nevo et al. (2020) to support this point.
Nevo, O., M. H. Schmitt, M. Ayasse, and K. Valenta. 2020. Sweet tooth: Elephants detect fruit sugar levels based on scent alone. Ecology and Evolution 10:11399-11407.
4. Page 3, line 139 –– “bipedal apes”. Be careful here. The earliest of evidence for bipedalism is approx. 7 Mya. I think you mean to say “congruent with increasing terrestrialization among African apes”
5. Page 4, lines 180-191 –– I think it would serve your interests here to say a little more about Ficus mucuso, and how the absence of outward color changes requires chimpanzees to engage in elaborate sensory assessments, including active olfaction. Sugiyama (1968) described the selection of F. mucuso figs by chimpanzees as “complicated” and “careful.” Dominy et al. (2016) described similar assessments with a different fig species in Kibale. My point is that most readers won’t understand why some of these figs are being rejected, or that smelling them is instrumental to efficient selection.
Sugiyama, Y. 1968. Social organization of chimpanzees in the Budongo Forest, Uganda. Primates 9:225-258.
Dominy, N. J., J. D. Yeakel, U. Bhat, L. Ramsden, R. W. Wrangham, and P. W. Lucas. 2016. How chimpanzees integrate sensory information to select figs. Interface Focus 6:20160001.
6. Page 5, lines 199-202 –– I get that the authors are planning to publish a comparative data set, but I think it would be ok to have a table focused squarely on F. mucuso at different stages of development. I must admit to feeling disappointed by the absence of empirical findings.
Author Response
Responses to Reviewer 2:
1. We now include this reference and modify the text accordingly.
2. So modified.
3. Ethanol was not demonstrated to be the sole olfactory cue in this otherwise suggestive study, so we prefer not to includ this (non-primate) example.
4. So modified.
5. We have added text on the relative paucity of color changes and the utility of olfactory cues in identifying ripe figs, and include the Dominy et al. 2016 reference to this end.
6. This review was not intended to provide detailed methodology or explicit empirical findings for an ongoing study, but rather to indicate that our results (in progress) for chimpanzee-consumed fruit are consistent with those already published for other tropical fruits, and to provide initial estimates as to likely ethanol dosage for our nearest living relatives under natural circumstances.
Reviewer 3 Report
This short review paper addresses research undertaken since the “drunken monkey” hypothesis was proposed circa 20 years ago, providing additional contextual results from recent field work in Uganda. Within the scope of the special issue “The Impact of Alcoholic Beverages on Human Health,” the paper is a good introductory piece for the volume and will be of interest to the readership of this journal. Overall, only very minor revisions are necessary (noted below); nevertheless, I recommend Section 3 be expanded to better review epigenetic factors associated with ethanol consumption at both an evolutionary and ontogenetic time scales rather than confounding the two.
Minor points:
- Page 1, lines 31-34: in standard archeological terminology, crop domestication is associated with the Neolithic and not the Paleolithic time period. Either (i) the correct time period should be noted or (ii) the Paleolithic-Neolithic transition should be referenced.
- Page 2, line 62: remove first instance of “also” in “…also must also…”
- Page 3, lines 106-108: move “…among wild primates…” clause after “…a recent survey…” in order to avoid confusion from grammatical structure
- Page 5, line 194: specify “IR” acronym (infrared gas?)
- Page 5, lines 200-201: clarify what percent refers to (weight ethanol to total fruit weight ratio?)
- Page 5, line 235: clarify 1400 m reference (i.e. 1400 masl)
Author Response
Responses to Reviewer 3:
Major point: We have reorganized the text to decouple discussion ontogenetic from evolutionary effects, and to better move from broad interspecific to (human) intraspecific studies, but also now emphasize the lack of relevant epigenetic results from non-laboratory systems or taxa, so have not substantially expanded that associated section.
Minor points:
1. So nuanced textually.
2. So deleted.
3. So clarified.
4. So specified.
5. So clarified.
6. So clarified.
Round 2
Reviewer 1 Report
The manuscript is ready to be published.